# Detection of Zika and dengue viruses in wild-caught mosquitoes collected during field surveillance in an environmental protection area in São Paulo, Brazil

Karolina Morales Barrio-Nuevo[1], Mariana Sequetin Cunha[2], Adriana Luchs[3], Aristides Fernandes[1], Iray Maria Rocco[2], Luis Filipe Mucci[4], Renato Pereira de Souza[2], Antônio Ralph Medeiros-Sousa[1], Walter Ceretti-Junior[1], Mauro Toledo Marrelli [1] *

1 Epidemiology Department, School of Public Health, University of São Paulo, São Paulo, Brazil, 2 Vector-borne Disease Laboratory, Virology Center, Adolfo Lutz Institute, São Paulo, Brazil, 3 Enteric Disease Laboratory, Virology Center, Adolfo Lutz Institute, São Paulo, Brazil, 4 Superintendency for the Control of Endemic Diseases, State Health Department, São Paulo, Brazil

* mmarelli@usp.br

**Data Availability Statement:** All data generated or analyzed during this study are included in the manuscript.

## Abstract

Species of the genus *Flavivirus* are widespread in Brazil and are a major public health concern. The country's largest city, São Paulo, is in a highly urbanized area with a few forest fragments which are commonly used for recreation. These can be considered to present a potential risk of flavivirus transmission to humans as they are home simultaneously to vertebrate hosts and mosquitoes that are potential flavivirus vectors. The aim of this study was to conduct flavivirus surveillance in field-collected mosquitoes in the Capivari-Monos Environmental Protection Area (EPA) and identify the flavivirus species by sequence analysis in flavivirus IFA-positive pools. Monthly mosquito collections were carried out from March 2016 to April 2017 with $CO_2$-baited CDC light traps. Specimens were identified morphologically and grouped in pools of up to 10 individuals according to their taxonomic category. A total of 260 pools of non-engorged females were inoculated into C6/36 cell culture, and the cell suspensions were analyzed by indirect immunofluorescence assay (IFA) after the incubation period. IFA-positive pools were tested by qRT-PCR with genus-specific primers targeting the flavivirus NS5 gene to confirm IFA-positive results and sequenced to identify the species. *Anopheles cruzii* (19.5%) and *Wyeomyia confusa* (15.3%) were the most frequent vector species collected. IFA was positive for flaviviruses in 2.3% (6/260) of the sample pools. This was confirmed by qRT-PCR in five pools (83.3%). All five flavivirus-positive pools were successfully sequenced and the species identified. DENV serotype 2 (DENV-2) was detected in *Culex* spp. and *Culex vaxus* pools, while ZIKV was identified in *An. cruzii*, *Limatus durhamii* and *Wy. confusa* pools. To the best of our knowledge, detection of flavivirus species of medical importance has never previously been reported in these species of wild-caught mosquitoes. The finding of DENV-2 and ZIKV circulating in wild mosquitoes suggests the existence of an enzootic cycle in the area. In-depth studies of DENV-2 and ZIKV, including investigation of mosquito infection, vector competence

**Funding:** This work was funded by the São Paulo Research Foundation (FAPESP: grants nos. 2014/50444-5 and 2014/10919-4) and the National Council for Scientific and Technological Development, Brazil (CNPq: grant no. 301466/2015-7).

**Competing interests:** The authors declare that they have no competing interests.

and infection in sylvatic hosts, are needed to shed light on the transmission dynamics of these important viruses and the potential risk of future outbreaks of DENV-2 and ZIKV infections in the region.

## Introduction

Over 700,000 deaths worldwide every year are caused by infections transmitted by blood-feeding arthropods, accounting for 17% of all infectious diseases [1, 2]. Examples of these arthropods include mosquitoes (Diptera: Culicidae), which are competent vectors for viruses of great epidemiological importance, as seen in recent major outbreaks and epidemics of Chikungunya-virus (CHIKV), Dengue-virus (DENV), Zika-virus (ZIKV) and Yellow Fever-virus (YFV) infections in Brazil [3, 4]. Arboviruses (arthropod-borne viruses) are found worldwide, and their emergence and reemergence usually manifest as infections with mild to severe clinical symptoms in humans and domestic animals, occasionally progressing to death. These diseases therefore have a considerable impact on public health and the economy of the affected region [5–7].

Among the arboviruses circulating in Brazil, members of genus *Flavivirus* (family *Flaviviridae*) are noteworthy as they are etiological agents of some of the most common viral infections and diseases in humans. In addition to DENV, ZIKV and YFV, several other flaviviruses of medical importance have been isolated in Brazil, including Bussuquara virus (BUSV), Cacicaporé virus (CPCV), Rocio virus (ROCV), Iguape virus (IGUV), Ilhéus virus (ILHV) and Saint Louis encephalitis virus (SLEV) [8–11]. Dengue is a reemerging disease in Brazil, with over 2 million confirmed cases and 702 recent deaths [12]. ZIKV has gained global attention as its geographic distribution has expanded dramatically from equatorial Africa and Asia to the Pacific Islands, South America and the Caribbean, causing many cases of neurological disorders and neonatal malformations [13–15].

Prevention and control of arboviral diseases require continuous surveillance and vector control measures. Investment in appropriate integrated surveillance measures should therefore be a priority in Brazil, especially considering the size of the country's population. Integrated surveillance, which covers epidemiological, entomological, sanitary and laboratory-based surveillance, is essential for early detection of epidemics and for rapid, effective control measures [16]. Donalisio et al. [16] stress that investment in epidemiological, virological, vector and epizootic surveillance measures should be a priority in Brazil. In the absence of specific treatment and an effective vaccine, ongoing entomological and epidemiological surveillance should be strengthened and integrated to control and prevent these arboviral diseases [3].

The Capivari-Monos Environmental Protection Area (EPA), in the south of the city of São Paulo, Brazil, is a forest remnant close to urban areas (an urban area). Previous studies in the Atlantic Forest in Brazil have shown that forest fragments offer favorable conditions for mosquitoes that are vectors of viruses to shelter and proliferate [17–19]. As circulating flaviviruses in city forest fragments can potentially cause disease outbreaks by infecting visitors and residents in neighboring areas, and given the lack of information about flavivirus-infected mosquitoes in parks in the city of São Paulo, the aim of this study was to conduct flavivirus surveillance in field-collected mosquitoes in the Capivari-Monos EPA and identify the virus species by sequence analysis.

## Material and methods

### Study area and mosquito sampling

The study was approved by the Ethics Committee of the University of São Paulo, Brazil, and collection permits were obtained in the Brazilian Biodiversity Authorization and Information System (SISBIO: Number: 44740–3).

Surveillance was conducted in the Capivari-Monos EPA, an area extending over 251 km$^2$ in the Atlantic Forest in the extreme south of the city of São Paulo where sustainable use of natural resources is practiced (Fig 1). Representing around one-sixth of the area of the whole municipality and bordering on the Serra do Mar State Park, Capivari-Monos EPA extends over the first hills and ocean slopes toward the upper reaches of Serra do Mar range, at altitudes varying from 740 to 800 m above sea level. It has a super-humid, tropical, ocean climate with average annual temperatures of around 19˚C and rainfall of between 1,600 and 2,200 mm. The vegetation is dense, tropical, montainous forest made up of Atlantic Forest remnants with different degrees of conservation, varying from well-conserved original forest to areas that have undergone a process of regeneration since 1950 and others that have been degraded recently as a result of rural and especially urban expansion. The district of Engenheiro Marsilac, in the Parelheiros region, lies in the EPA and has around 10,000 inhabitants, most of whom are low-income settlers. The population density is approximately 41 inhabitants per km$^2$ [20, 21], and the area is home to the non-human primate (NHP) *Alouatta guariba clamitans* [22].

Mosquito collections were carried out monthly from March 2016 to April 2017 in forested areas in Engenheiro Marsilac with different levels of anthropogenic intervention. Specimens were collected in (1) Embura—a village surrounded by small farms and the EPA forest (23˚ 53.036′ S/46˚ 44.485′ W); (2) Marsilac—a village surrounded by the EPA forest and near a railway line (23˚ 54.395′ S/46˚ 42.486′ W); (3) Transition zone—private property near Marsilac village constituting a transitional area between a rural environment and the EPA forest (23˚ 54.556′ S/46˚ 42.167′ W); and (4) Wild area—private property in the EPA forest next to a waterfall with a visitation area (23˚ 56.378′ S/46˚ 41.659′ W) (Fig 1). In each collection area two $CO_2$-baited CDC light traps [23] with Lurex3™ were installed, one in the tree canopy (>10 m) and another at ground level. All the traps were set up early in the afternoon and removed after 18 hours of exposure.

Specimens were carried alive to the Entomology in Public Health Laboratory at the School of Public Health, University of São Paulo (LESP/FSP/USP), where they were morphologically identified on a specially designed chill table with a stereo microscope and the dichotomous keys described by Consoli & Lourenço-de-Oliveira [24] and Forattini [25]. Non-engorged females were grouped in pools of up to 10 individuals according to their taxonomic category and place and date of collection, giving a total of 260 pools. The pools were then transported in dry ice to the Vector-borne Diseases Laboratory, Adolfo Lutz Institute, and stored at -70˚C until use.

### Detection of flaviviruses

The 260 pools were macerated individually in tubes containing 1 mL of 1.8% bovine albumin and antibiotics (100 units/mL of penicillin and 100 μL/mL of streptomycin) and centrifuged at 2,500 rpm for 10 min. The supernatant was stored at -70˚C until inoculation into C6/36 cell culture (*Aedes albopictus* clone). The cultures were incubated for nine days at 28˚C with L-15 medium containing 2% FBS, penicillin (100 units/mL) and streptomycin (100 μg/mL). After the incubation period, the cells were then scraped off the tube and the cell suspensions were spotted onto a glass slide, which was air dried, fixed with acetone and used for an Indirect

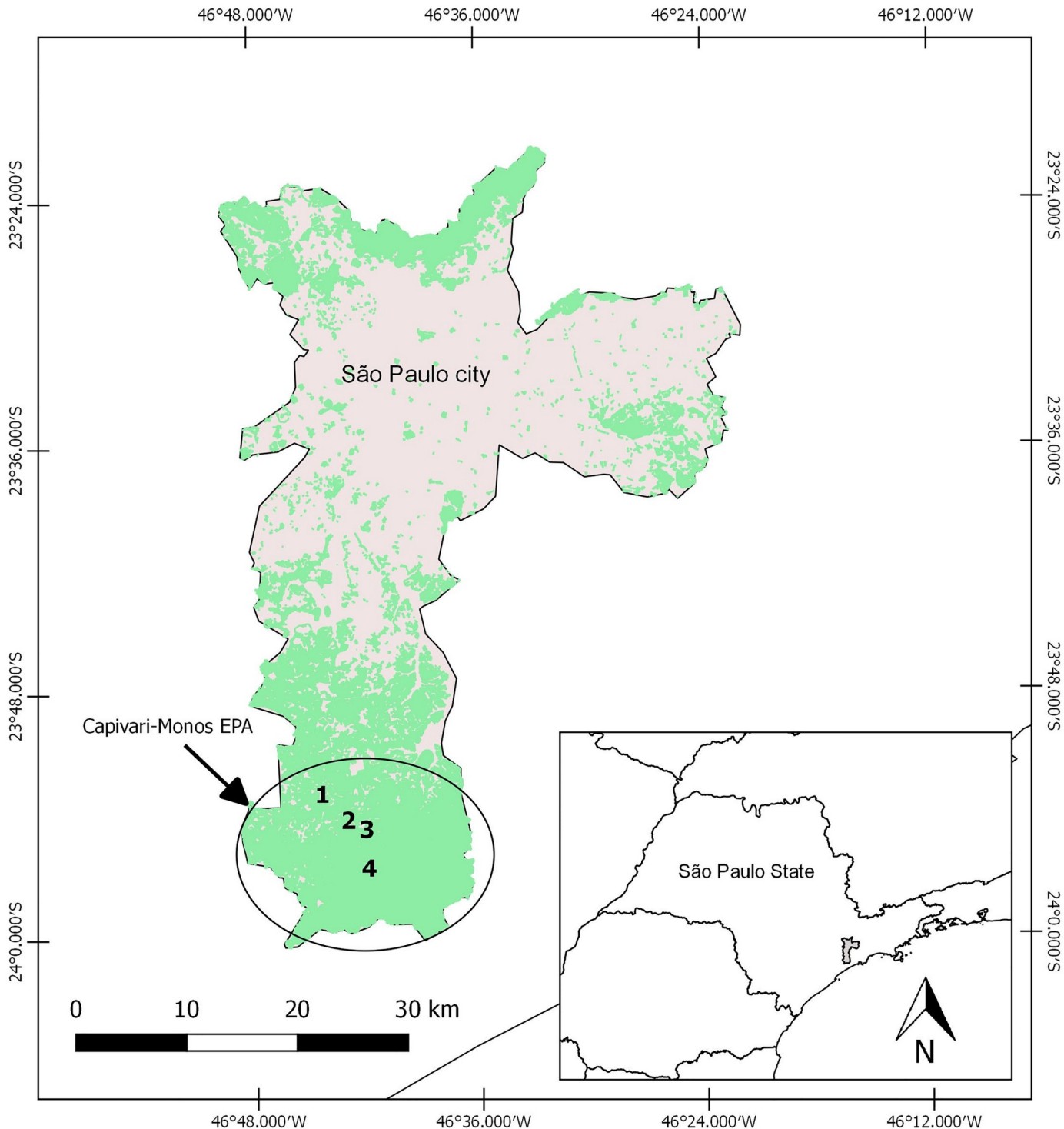

**Fig 1. Capivari-Monos EPA in the extreme south of the city of São Paulo, SP, Brazil.** Collection sites are numbered as follows: (1) Embura village, (2) Marsilac village, (3) Transition zone, (4) Wild area. Green represents dense tropical forest. Grey represents areas where there is human activity (villages, roads and rural properties). The map was created using QGIS v2.18.9 (http://www.qgis.org).

Immunofluorescence Assay (IFA) with in-house anti-flavivirus polyclonal antibody and FITC-labeled anti-mouse IgG (whole molecule) antibody (Sigma-Aldrich, St. Louis, MO, USA) [26, 27]. The slides were examined under an epifluorescence microscope.

## Identification of flavivirus species

Samples positive for flaviviruses in IFA were analyzed by real-time reverse transcription polymerase chain reaction (qRT-PCR) to confirm the result and sequenced to identify the species. Viral RNA was isolated from 140 μL aliquots of supernatants from IFA-positive cell cultures which had been stored at -70˚C using the QIAamp$^{®}$ Viral RNA Mini Kit (Qiagen, Valencia, CA, USA) according to the manufacturer's instructions and tested using the pan-flavi qRT-PCR assay described previously by Patel et al. [28]. Positive qRT-PCR amplicons (~260 bp) were sequenced directly with the primers Flavi S and Flavi AS2 [28] and BigDye Kit v3.1 (Applied Biosystems, Inc., Foster City, CA, USA). Dye-labeled products were sequenced on an ABI 3130 sequencer (Applied Biosystems, Inc., Foster city, CA, USA). Chromatograms were edited manually with Sequencer 4.7 software and screened with the Basic Local Alignment Search Tool (BLASTn and BLASTx). Sequences generated in this way and a set of cognate sequences of DENV and ZIKV available in GenBank (S1 Table) were aligned with the BioEdit sequence alignment editor (version 7.0.5.2) [29]. Neighbor Joining (NJ) trees were constructed with the Kimura 2-parameter model in MEGA 6.0 and 1,000 bootstrap replicates [30]. Reference ZIKV and DENV sequences were added to the corresponding tree so that species identity could be confirmed. The nucleotide (nt) sequences were deposited in GenBank under accession numbers MK134005, MK134006, MK371391, MK371392 and MK371393.

## Results

In total, 878 specimens of mosquitoes belonging to 37 taxa (11 genera) were sampled, of which 99.8% were female and 0.2% male. Most of the specimens were collected in the canopy (54.1%), 41.1% at ground level and for 4.8% information on stratification was not available. The species *Anopheles (Kerteszia) cruzii*, (171 specimens), *Wyeomyia* (*Prosopolepis*) *confusa* (134), *Culex* (*Culex*) spp. (109), *Limatus durhamii* (61), *Wyeomyia* (*Phoniomyia*) *theobaldi* (58) and *Culex* (*Melanoconion*) *vaxus* (36) were the most common species collected. More mosquitoes (61.4%) were collected in the wild area than in the transition zone (14.9%), Embura (13.1%) and Marsilac (8.7%) (S2 Table).

Flaviviruses were detected by IFA in 2.3% (6/260) of the sample pools. However, only five IFA-positive pools of non-engorged females (83.3%) were confirmed by qRT-PCR. All five positive pools were successfully sequenced, and the species identified with an ~200 bp long NS5 fragment. DENV serotype 2 (DENV2) was detected in *Cx*. spp. and *Cx. vaxus* pools, while ZIKV was detected in *An. cruzii*, *Li. durhamii* and *Wy. confusa* pools. The flavivirus-positive pools came from two distinct areas in the EPA: the wild area and the transition zone (Table 1).

The NJ trees constructed include the isolates sequenced in this study (in bold) and reference sequences from GenBank. The trees confirm the classification of Flavi-143 and Flavi-168 as DENV-2 since the sequences formed well-supported monophyletic groups with their corresponding reference isolates (Fig 2). The two isolates in the present study share an nt identity of 94.1%. Genetic analysis of the partial NS5 gene revealed that the DENV-2 isolates identified here are more closely related to those isolated in humans in Cuba in 1981, China in 2015 and Papua New Guinea in 1944 (93.3–97.6% identity) than to the Brazilian human BID-V2377 isolated in 2000 (88.6–93.7% identity). Genetic analysis of the partial NS5 gene sequences revealed that the three Brazilian ZIKV isolates exhibited high nt identity (97.0–98.1%) with other ZIKV isolates detected in the Americas since 2015, including isolates from humans,

**Table 1. Flavivirus-positive samples detected in pools of mosquitoes from the Capivari-Monos EPA, São Paulo, Brazil, 2016–2017.**

| Pool code (Sample) | Collection date | Location | Taxon | Number of mosquitoes per pool (N) | Forest stratum | Flavivirus species | Isolate | GenBank accession no. |
|---|---|---|---|---|---|---|---|---|
| 168 14P-F | October 2016 | Wild area | *Cx.* spp. | 6 | Canopy | DENV-2 | Flavi-168-DENV-2 | MK134006 |
| 143 9P-C | February 2017 | Wild area | *Cx. vaxus* | 4 | Canopy | DENV-2 | Flavi-143-DENV-2 | MK134005 |
| 148 11P-A | February 2017 | Transition zone | *Li. durhamii* | 6 | Ground | ZIKV | Flavi-148- ZIKV | MK371391 |
| 157 14P-A2 | October 2016 | Wild area | *An. cruzii* | 10 | Canopy | ZIKV | Flavi-157- ZIKV | MK371393 |
| 150 12P-A | February 2017 | Transition zone | *Wy. confusa* | 2 | Canopy | ZIKV | Flavi-150- ZIKV | MK371392 |

Descriptions of flavivirus-positive pools showing pool code, collection date, collection site, taxon, number of specimens, stratification, viral RNA identified, isolate and GenBank accession number.

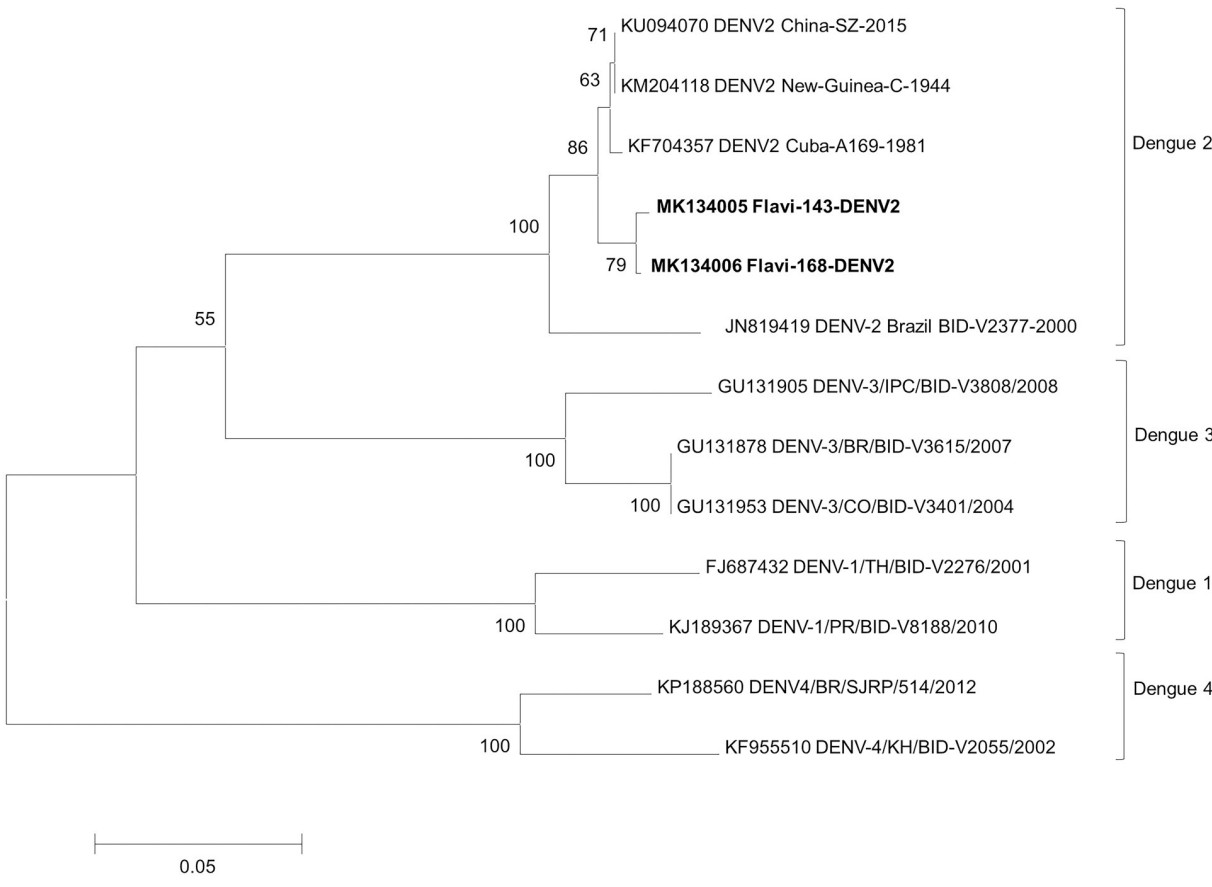

**Fig 2. Neighbor-Joining (NJ) phylogenetic tree of partial NS5 gene sequences (generated in MEGA 6.0) of DENV-2 isolates detected in *Cx. ssp and Cx. vaxus* in the Capivari-Monos EPA, São Paulo, Brazil, 2016–2017.** Reference DENV-1, DENV-2, DENV-3 and DENV-4 isolates were retrieved from GenBank. The species and accession number of each isolate are indicated. The scale indicates the number of divergent nucleotide residues. Percentage bootstrap values are shown at the branch node.

*Aedes* mosquitoes and NHPs, and 99.6% nt identity with each other. The three ZIKV lineages (America, Asia and Africa) can be clearly distinguished in Fig 3.

## Discussion

The ecological networks that connect flaviviruses to their vectors and hosts are varied, complex and poorly understood. However, the ecology and epidemiology of these viruses can be better understood through mosquito-based flavivirus surveillance [31]. In the present study, we have described the detection of DENV-2 and ZIKV in *Cx.* spp, *Cx. vaxus*, *Li. durhamii*, *An. cruzii* and *Wy. confusa* captured in the Capivari-Monos EPA during the period 2016 to 2017. DENV-2 and ZIKV are known to have a great impact on human health [32–34], particularly in urban centers, and several studies have reported the finding of mosquitoes that are considered important flavivirus vectors in fragments of the Atlantic Forest in various areas of the city [18, 35–37] and state [4, 38, 39] of São Paulo. To the best of our knowledge, the detection of flavivirus species of medical importance has never been previously reported in wild-caught mosquitoes in the area in which the present study was carried out.

*Culex vaxus* appears to retain behavioral characteristics typical of wild mosquitoes as it has been shown to adapt poorly to areas with reduced forest [40]. Although information about this species is still quite scarce, it is known to have eclectic feeding habits and to be able to participate in sylvatic arbovirus cycles [24, 25]. Furthermore, viruses have been detected in some species of the genus *Culex* collected in wild environments [24]. *Culex* mosquitoes are vectors of a

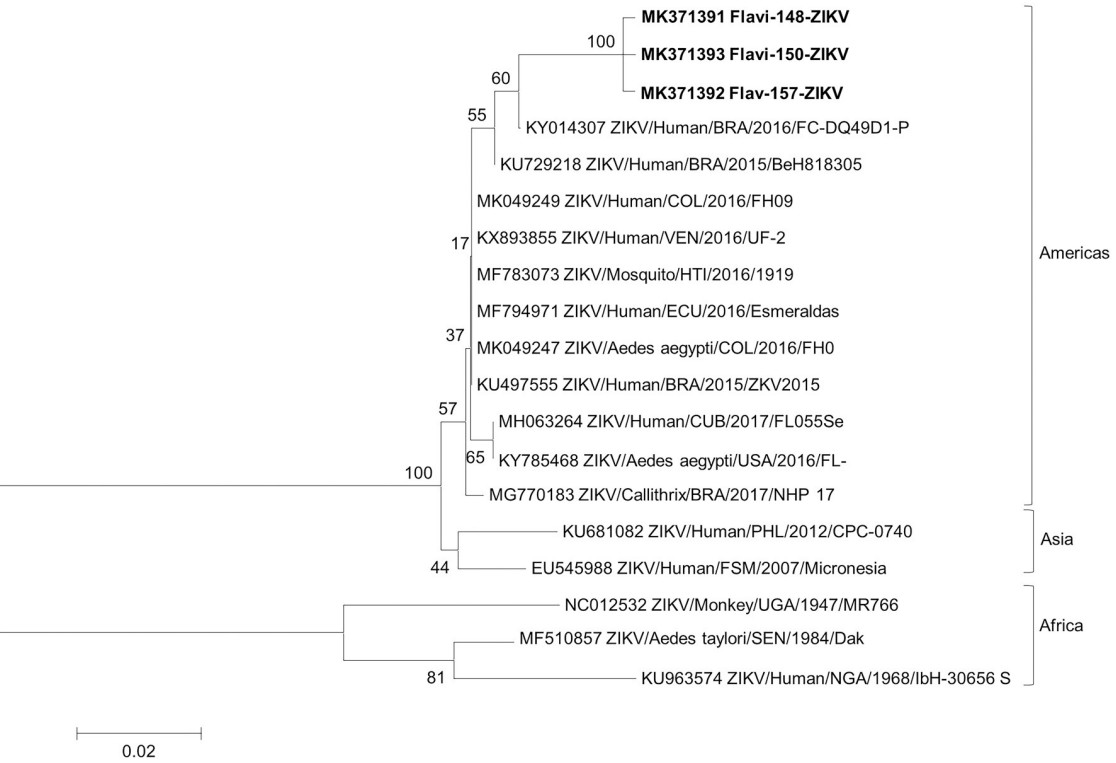

**Fig 3. Neighbor-Joining (NJ) phylogenetic tree (generated in MEGA 6.0) of partial NS5 gene sequences of ZIKV isolated from *Li. durhamii*, *An. cruzii* and *Wy. confusa* in the Capivari-Monos EPA, São Paulo, Brazil, 2016–2017.** Reference ZIKV sequences were retrieved from GenBank. The species and accession number of each isolate are indicated. The scale indicates the number of divergent nucleotide residues. Percentage bootstrap values are shown at the branch nodes.

considerable number of flaviviruses that occur in Brazil, such as West Nile virus, Saint Louis Encephalitis virus and Ilheus virus [24, 41, 42]. Although some studies have demonstrated replication of DENV-2 in species of the genus *Culex*, the general consensus is that *Cx.* spp. are not biological vectors of dengue viruses [43]. However, Guedes et al [44] demonstrated recently that *Cx. quinquefasciatus* is a competent vector for ZIKV.

*Anopheles cruzii* is generally restricted to the Brazilian coast and has a distribution similar to the original distribution of the Atlantic Forest [25]. The high abundance of this species is directly related to the high availability of natural breeding sites but can also be attributed to the species' opportunistic behavior and eclectic feeding habits [24, 25]. *An. cruzii* is commonly associated with *Plasmodium* transmission in humans and simians in the Atlantic Forest. Its feeding source includes NHPs [45–47] and although usually associated with malaria, the species has been found infected with arboviruses. Epelboin et al. [48] pointed out that *An. cruzii* is also involved in O'nyong-nyong virus (*Alphavirus*) transmission, and Cunha et al. [42] recently reported the finding of *An. cruzii* naturally infected with the Iguape virus (genus *Flavivirus*) in field samples collected in Juquitiba, São Paulo, in 1994. In addition, two species of the genus *Anopheles*, *An. coustani* and *An. gambiae*, were found naturally infected with ZIKV in Africa [48].

*Wyeomya confusa* and *Li. durhamii* belong to tribe Sabethini and have close phylogenetic relationships [24, 25]. The former is an opportunistic species and frequently bites humans [25]. Although it is usually sylvatic, *Wy. confusa* was collected here in an environment with greater anthropic interventions (Embura) and in a transition zone. While this species has already been found infected with arboviruses in wild environments, there is virtually no information about its medical importance [25]. *Li. durhamii* is the species of tribe Sabethini best adapted to anthropic environments [24] and there have been reports of this mosquito carrying Guama, Tucunduba and Maguari viruses (orthobunyaviruses) [49]. Intriguingly, ZIKV was detected in these three different species of sylvatic mosquitoes in the Capivari-Monos EPA.

One of the criteria used to determine whether sylvatic cycles of arboviruses are occurring is the presence of a sufficient number of susceptible NHPs and competent mosquitoes that feed on NHPs [50]. The finding of arboviruses in remote forest-dwelling mosquitoes is considered reliable evidence of a sylvatic cycle [32, 50]. Flaviviruses with known sylvatic cycles include DENV serotypes 1–4, ZIKV in parts of Africa and Asia [15, 48, 51–55] and YFV in South America and Africa [56]. In Brazil, the YFV cycle is sustained by NHPs and *Haemagogus* spp. mosquitoes [56]. As for a sylvatic cycle of ZIKV, this virus was detected in periurban *Callithrix* monkeys in the state of São Paulo during a human outbreak [57], but to date no field studies on wild-caught mosquitoes have detected ZIKV, and a study using colonized *Sabethes cyaneus* mosquitoes showed only a low vector competence for this virus [58].

The detection of DENV-2 and ZIKV in these sylvatic mosquitoes must be treated with caution. Identification of DENV-2 and ZIKV in sylvatic mosquitoes may not necessarily be associated with natural infection and may not indicate that these mosquitoes play a role in the transmission of flaviviruses as the virus detected could have originated from blood remnants in mosquitoes that had blood fed a few days before being caught. To better elucidate the existence of a sylvatic cycle of DENV-2 and ZIKV in Brazil, further studies on the infectivity and vector competence of wild-caught species of mosquitoes [24, 25, 44] are required, as well as investigations into the presence of arboviruses in different species of Brazilian NHPs.

The frequency of species of the genus *Flavivirus* detected in this study (2.6%) was similar to that observed in other studies in Italy (2.3%) [59], China (5.1%) [60] and Brazil (2.8%; 4.0%) [19, 61]. Cunha et al. [4] found a frequency of 0.29% of mosquitoes infected with YFV during the 2016–2017 outbreak in the state of São Paulo. Detection of flavivirus RNA in mosquitoes is not an easy task as detection of viruses in mosquitoes requires that large samples of specimens

be screened. Gu et al. [62] showed that to detect arboviruses in mosquito populations with low levels of infection requires samples with more than 1,600 individuals for a high probability (0.8) of detection. Sample size is thus a key determinant of arbovirus detection in mosquitoes, making intensified mosquito surveillance essential to detect arboviral activity [62].

Phylogenetic analysis based on the partial NS5 gene showed that the Brazilian DENV-2 and ZIKV sequences clustered together with sequences from distinct continents, suggesting that the flaviviruses circulating in the Capivari-Monos EPA are genetically related to those circulating worldwide. While there is no evidence that a particular variant is spreading in the study area, it should be stressed that the phylogenetic tree in the present study was generated with short sequences (~200 bp). Phylogenetic reconstructions based on longer genomic sequences might provide more robust, informative data.

Although DENV-2 and ZIKV are considered urban and periurban arboviruses [13, 14, 34, 63], in the present study they were identified in a conserved wild area. In fact, this is the first study to have detected DENV-2 and ZIKV in mosquitoes captured in a conservation unit in the city of São Paulo. The finding of DENV-2 and ZIKV circulating in the Capivari-Monos EPA suggests an enzootic cycle in the area. *Aedes aegypti*, the main vector of these two flaviviruses of medical importance in the Americas [16, 64], was not found in the study area, corroborating previous findings [17]. Nevertheless, the Parelheiros region (where the study area is) has registered autochthonous dengue cases in the last few years, and these have spread progressively towards Engenheiro Marsilac [17]. Other studies carried out in urban parks in São Paulo also failed to find *Ae. aegypti* [19, 65, 66], highlighting the anthropophilic nature of this species. A search for DENV and ZIKV in NHPs together with mosquito and virological field surveillance are needed to investigate the potential role of mosquito species circulating in EPAs and whether urban arboviruses can establish an enzootic cycle in the city of São Paulo.

## Conclusion

DENV-2 and ZIKV were found in *Culex*, *Anopheles*, *Limatus* and *Wyeomyia* mosquitoes in the Capivari-Monos EPA, confirming the presence and activity of these two potentially medically important flaviviruses in this conservation unit. Flavivirus surveillance in field-caught mosquitoes is essential for continuous monitoring of virus activity and for an understanding of the ecology and epidemiology of species of this genus. In addition, in-depth studies of DENV-2 and ZIKV, including vector competence and molecular studies, are needed to shed light on the transmission dynamics of these two arboviruses and the potential risk of future epidemic outbreaks in the study region.

## Supporting information

**S1 Table. Reference sequences of dengue and Zika viruses from GenBank.** Sequences of dengue and Zika viruses from GenBank aligned to construct a phylogenetic tree by country of origin, isolate, year, origin of isolated material, genome sequence and GenBank accession number.
(DOCX)

**S2 Table. Mosquito species collected in the Capivari-Monos Environmental Protection Area (EPA).** The table shows the number of individuals at each study point, forest stratum, sex and number of pools. Mosquitoes collected from March 2016 to April 2017.
(DOCX)

## Acknowledgments

We would like to express our gratitude to the following staff of the Superintendency for the Control of Endemic Diseases, São Paulo Zoonosis Control Center, and the School of Public Health, São Paulo University: Dr. Ana Maria Ribeiro de Castro Duarte, João Carlos do Nascimento, Paulo Frugoli dos Santos, Luis Milton Bonafé, Antônio Waldomiro de Oliveira, Laércio Molinari, Gabriel Marcelino Neto, Luiz Sposito Jr, Renildo Souza Teixeira, Daniel Pagotto Vendrami, Laura Cristina Multini, Gabriela Cristina de Carvalho, Ramon Wilk da Silva, Rafael de Oliveira Christe, Eduardo Evangelista de Souza, Amanda Alves Camargo and Ana Leticia da Silva de Souza.

## Author Contributions

**Conceptualization:** Karolina Morales Barrio-Nuevo, Renato Pereira de Souza, Mauro Toledo Marrelli.

**Data curation:** Mariana Sequetin Cunha, Adriana Luchs, Walter Ceretti-Junior.

**Formal analysis:** Mariana Sequetin Cunha, Adriana Luchs, Aristides Fernandes, Renato Pereira de Souza, Antônio Ralph Medeiros-Sousa, Walter Ceretti-Junior, Mauro Toledo Marrelli.

**Funding acquisition:** Mauro Toledo Marrelli.

**Investigation:** Karolina Morales Barrio-Nuevo, Mariana Sequetin Cunha, Adriana Luchs, Aristides Fernandes, Iray Maria Rocco, Luis Filipe Mucci, Renato Pereira de Souza, Antônio Ralph Medeiros-Sousa, Walter Ceretti-Junior.

**Methodology:** Mariana Sequetin Cunha, Adriana Luchs, Iray Maria Rocco, Luis Filipe Mucci, Renato Pereira de Souza, Antônio Ralph Medeiros-Sousa.

**Resources:** Mauro Toledo Marrelli.

**Supervision:** Mariana Sequetin Cunha, Renato Pereira de Souza, Mauro Toledo Marrelli.

**Validation:** Mariana Sequetin Cunha, Renato Pereira de Souza.

**Writing – original draft:** Karolina Morales Barrio-Nuevo, Mauro Toledo Marrelli.

**Writing – review & editing:** Mariana Sequetin Cunha, Adriana Luchs, Iray Maria Rocco, Antônio Ralph Medeiros-Sousa, Walter Ceretti-Junior, Mauro Toledo Marrelli.

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
