## [Decision Letter · Decision Letter 0]

9 Jan 2020

PONE-D-19-34539

Zika and dengue viruses infecting wild-caught mosquitoes in an environmental protection area in Brazil

PLOS ONE

Dear Dr Marrelli,

Thank you for submitting your manuscript to PLOS ONE. After careful consideration, we feel that it has merit but does not fully meet PLOS ONE’s publication criteria as it currently stands. Therefore, we invite you to submit a revised version of the manuscript that addresses the points raised during the review process.

We would appreciate receiving your revised manuscript by Feb 23 2020 11:59PM. To enhance the reproducibility of your results, we recommend that if applicable you deposit your laboratory protocols in protocols.io, where a protocol can be assigned its own identifier (DOI) such that it can be cited independently in the future. For instructions see: http://journals.plos.org/plosone/s/submission-guidelines#loc-laboratory-protocols

We look forward to receiving your revised manuscript.

Kind regards,

George Dimopoulos, PhD MBA

Academic Editor

PLOS ONE

Journal Requirements:

Additional Editor Comments (if provided):

Reviewers' comments:

Reviewer's Responses to Questions

**Comments to the Author**

1. Is the manuscript technically sound, and do the data support the conclusions?

Reviewer #1: Partly

Reviewer #2: Yes

2. Has the statistical analysis been performed appropriately and rigorously? 

Reviewer #1: N/A

Reviewer #2: Yes

3. Have the authors made all data underlying the findings in their manuscript fully available?

Reviewer #1: Yes

Reviewer #2: Yes

4. Is the manuscript presented in an intelligible fashion and written in standard English?

Reviewer #1: Yes

Reviewer #2: Yes

5. Review Comments to the Author

Reviewer #1: Overall, I find this study to be sound and interesting, but not enough detail provided about the methods or the findings for it to be accepted without revisions. The authors describe several times that they identified mosquitoes "infected" with virus, but due to the lack of details about the IFA assay which appears to be the only assay used that might demonstrate infectivity, from just the data provided this statement is inaccurate. More information and data regarding the IFA must be provided, including relevant information around positive and negative controls, and how samples were determined to be positive, before samples could be suggested to be "infected". For instance, if the IFA is focus forming and the number of infectious virions in each mosquito pool could be quantified, this would add strength to the conclusion that the mosquitoes were "infected". Otherwise, the authors must rephrase and restructure their discussion to reflect the limited findings of virus RNA in inoculated cell culture samples. Please see attached document for additional comments.

Reviewer #2: Barrio-Nuevo and colleagues present a surveillance study of flaviviruses in a natural area in Sao Paolo. They report isolation and partial sequencing of ZIKV and DENV from this area, which suggests sylvatic perpetuation of these agents in this region. The work appears to have been conducted and analyzed in a technically acceptable manner and the conclusions are justified by the data. Moreover, this work contributes to our knowledge about flavivirus lifecycles in Brazil.

6. PLOS authors have the option to publish the peer review history of their article (what does this mean?). If published, this will include your full peer review and any attached files.

Reviewer #1: No

Reviewer #2: No

---

## [Author Response · Author response to Decision Letter 0]

20 Feb 2020

To the Editor 

Plos One

Dear Dr. Dimopoulos

Thank you for reviewing our manuscript. We have made substantial changes to the scientific note to address the points made by the reviewers. We believe the changes proposed by the reviewers have improved the article significantly. 

Sincerely,

Mauro Marrelli

Answers: Reviewer #1

 We are grateful to the reviewer for his/her very useful comments. All the suggestions and criticisms have been addressed.

Query 1

Line 44 – “Mosquitoes are competent vectors for viruses…”

Query 2

Line 46 – no hyphens between virus names

Query 3

Line 67 – “a priority” not “priorities”

Answer: We thank the reviewer for pointing out these mistakes and have made the necessary corrections to the manuscript.

Query 4

Line 127 – The Gubler et al reference contains many different types of virus detection and therefore the exact methods you used are not clear from that reference. Please describe the methods of your IFA assay.

Answer: A description of the methods used in the IFA has been added.

Query 5

Line 127 – Is this a focus-forming assay? Is it possible to quantify the number of foci and determine how many infectious virus particles were in the mosquito pool? Were positive controls used to compare to positive results? How was “positive” determined for the IFA? What about negative controls for antibody background staining? Please clarify these points, it will improve confidence in the findings of this study.

Answer: The IFA is not a focus-forming assay. After inoculation of the C6/36 cells with the mosquito pools, the virus replicated (this is why we kept it for 9 days, which is the mean time for dengue virus replication in Vero cell lines). In the IFA the cytoplasm of the cells was fluorescent, showing that the virus was able to replicate in this cell line and was viable, confirming viral isolation. We used the 17DD Yellow Fever virus as positive control, and a non-inoculated tube as negative control. 

Query 6

Line 131 – What quantification method did you use to determine the amount of viral RNA in your qRT-PCR assay? What was the basis of your standard curve?

Answer: We did not quantify the amount of RNA. We followed the approach described in the original article by Patel et al. (2013), who determined that positive results are those with Ct values lower than 39.7. This information has been added in the text.

Query 7

Line 137 – are these primers the Flavi S and Flavi AS2 primers used for sequencing (Line 140)? If not please clarify which primers were used and provide sequences 

Answer: Yes, we used the same primers, as stated in lines 151-152.

"The qRT-PCR amplicons (~260bp) were directly sequenced with the BigDyeTM kit v3.1 (Applied Biosystems, Inc., Foster City, CA, USA) and Flavi S and Flavi AS2 primers [26].

Query 8

Line 162 – is this also how many were positive by the IFA? Meaning, all the IFA positives were also positive by qRT-PCR?

Answer: Yes. All isolates were positive by qRT-PCR.

Query 9

Line 162 – Since this is a qRT-PCR assay, please report the quantities of viral RNA found in the supernatants of the C6/36 cells. This will help determine if the virus from the inoculated mosquito pool was infectious and replicated in the C6/36 cells.

Answer: We did not quantify RNA in the supernatants of the C6/36 cells. The result is based on visual fluorescence of the cell cytoplasm where viruses replicated. As we were able to isolate Dengue and Zika viruses from the mosquito pools, we can assume that these viruses were viable, and therefore infectious.

Query 10

Line 162 – The qRT-PCR protocol you used, as described by Patel et al, is very sensitive (10-100 genome copies). Why not determine the presence of viral RNA directly in the mosquito pools themselves?

Answer: We used isolation of virus in C6/36 cells to avoid false positive results, which could be a problem using only the very sensitive qRT-PCR protocol described by Patel. A further advantage of isolation in relation to the qRT-PCR protocol is that it allowed us to prove that the virus is able to replicate in C6/36 cells. In addition, the virus-isolation approach we used is cheaper. 

Query 11

Line 206 – Technically this study doesn’t demonstrate that the mosquitoes are infected with the viruses, but that the mosquitoes contain infectious virions (if the IFA is indeed a focus forming assay). Just because there is no visible blood does not mean these mosquitoes have not blood fed recently and that some infectious virus could remain even if it isn’t infecting the mosquitoes themselves. Clarify language here and in other areas where “infected” is used

Answer: We thank the reviewer for pointing this out. Indeed, these viruses may have come from a recent human blood meal. However, we showed that the mosquitoes were infected as the virus was viable and could replicate in C6/36 cells.

Query 12

Line 258 – Guedes et al do not demonstrate ZIKV replication in Cx. quinquefasciatus – a closer look at the titers of virus RNA in those mosquitoes will show RNA levels for ZIKV remaining stagnant over time and do not increase at any point measured. Clarify language

Answer: We respectfully disagree with the reviewer. The article by Guedes et al. [Guedes DRD et al. Zika virus replication in the mosquito Culex quinquefasciatus in Brazil. Emerg Microbes Infect. 2017; doi: 10.1038/emi.2017.59.] does in fact demonstrate that ZIKV replicated and was detected in the salivary glands and saliva of artificially fed Cx. quinquefasciatus mosquitoes.

Query 13

Line 262 – Your study has absolutely not demonstrated this. You have found DENV/ZIKV RNA in cell culture samples that are positive for Flaviviruses by IFA. This indicates that pools of these mosquitoes may contain virions that can infect C6/36 cells, but does not indicate that these virions are infecting the mosquitoes in the pool, or that they would be disseminated in the mosquito saliva to new hosts. This sentence greatly misrepresents your findings and should be rewritten.

Answer:

We agree in part with the reviewer. Our results show that the virus was infecting the mosquitoes, was viable and could replicate in C6/36. But we agree that the vector competence needs to be clarified and have modified the sentence accordingly.

Query 14

Line 281 – Once again, you have not found these mosquitoes to be “infected”. Clarify language

Answer: As before, we agree in part with the reviewer and have therefore modified the sentence as requested. 

Query 15

Line 293 – I’m not sure one citation constitutes this mosquito being found “frequently… in natural and artificial breeding sites”. Possibly clarify this sentence or find more references

Answer: We agree and have modified the sentence accordingly.

Query 16

Line 295 – The sentence “This mosquito bites actively…” should have citations

Answer:

Thank you for pointing this out. We have accordingly added the reference.

---

## [Decision Letter · Decision Letter 1]

6 May 2020

PONE-D-19-34539R1

Zika and dengue viruses infecting wild-caught mosquitoes in an environmental protection area in Brazil

PLOS ONE

Dear Dr Marrelli,

Thank you for submitting your manuscript to PLOS ONE. After careful consideration, we feel that it has merit but does not fully meet PLOS ONE’s publication criteria as it currently stands. Therefore, we invite you to submit a revised version of the manuscript that addresses the points raised during the review process.

1) The authors need to make clear the methodology, including positive and negative controls. 

2) It is fundamental to determine if the mosquitoes were infected. The presence of the virus in mosquitoes extracts is not an indicator of infection.

3) The authors force the conclusion that these mosquitoes might be vectors, but there is a long way to go before this can be concluded.

We would appreciate receiving your revised manuscript by Jun 20 2020 11:59PM. To enhance the reproducibility of your results, we recommend that if applicable you deposit your laboratory protocols in protocols.io, where a protocol can be assigned its own identifier (DOI) such that it can be cited independently in the future. For instructions see: http://journals.plos.org/plosone/s/submission-guidelines#loc-laboratory-protocols

We look forward to receiving your revised manuscript.

Kind regards,

Humberto Lanz-Mendoza

Academic Editor

PLOS ONE

Reviewers' comments:

Reviewer's Responses to Questions

**Comments to the Author**

1. If the authors have adequately addressed your comments raised in a previous round of review and you feel that this manuscript is now acceptable for publication, you may indicate that here to bypass the “Comments to the Author” section, enter your conflict of interest statement in the “Confidential to Editor” section, and submit your "Accept" recommendation.

Reviewer #3: (No Response)

Reviewer #4: (No Response)

2. Is the manuscript technically sound, and do the data support the conclusions?

Reviewer #3: Partly

Reviewer #4: Partly

3. Has the statistical analysis been performed appropriately and rigorously? 

Reviewer #3: N/A

Reviewer #4: N/A

4. Have the authors made all data underlying the findings in their manuscript fully available?

Reviewer #3: Yes

Reviewer #4: Yes

5. Is the manuscript presented in an intelligible fashion and written in standard English?

Reviewer #3: Yes

Reviewer #4: Yes

6. Review Comments to the Author

Reviewer #3: The present study investigated the potential presence of arboviruses (more specifically flaviviruses) in wild caught mosquito species in a Environmental Reserve in Sao Paulo, Brazil. Although the findings are important for surveillance purposes, one cannot conclude that these mosquitoes species are vectors for these arboviruses. Although the authors claim they only used "non-engorged" females, there is a possibility that these mosquitoes had traces of blood in their midgut.

The authors emphasise in several instances that these mosquitoes are infected and therefore could be vectors of flaviviruses, which is not appropriate. This has to be worked throughout the manuscript.

Reviewer #4: The paper “Zika and dengue viruses infecting wild-caught mosquitoes in an environmental protection area in Brazil” describes the ZIKV and DENV infection in a C6/36 cell line, from inoculation with extracts pools (10) of wild collected mosquitoes in Brazil. Flavivirus detection was carried out by IFA and identification by qRT-PCR and sequencing. The results are interesting, but for this reviewer additional information is needed and some queries must be clarified.

Line 119. “Detection of flaviviruses” This methodology is 1/3 of the whole Materials and Methods section, but no results are provided (it is surprising because only five samples were positive). This reviewer consider it so important to the readers, mainly because the comparison between positive samples with positive and negative controls.

Line 131. “…primary antibody a polyclonal anti-St Louis encephalitis virus antibody...” I have the concern about if such antibody is able to detect any flavivirus (cross-reaction), if so, a reference should be provided (Gluber et al., used a monoclonal antibody against DENV).

Line 135. Why YFV was used as positive control? Were C6/36 infected with YFV and then performed the IFA? Again, the polyclonal anti-St Louis encephalitis virus antibody also recognize YFV (reference)? Was the IFA signal in YFV similar to problem samples?

Please clarify.

Line 138. “Positive flavivirus samples were analyzed...” Which positive samples (IFA)? Please clarify.

Line 166-167. Full genus name must be wrote at first time it appears if the main body text.

Line 178. Table 1. It is not clear in the table if N (number of specimens) correspond to the amount of mosquitoes (number) or number of analyzed pools. Mainly because in the first column, the key number looks to indicate single pools. If so, there are pools from 2 to 10 mosquitos. Please clarify; it is so important to interpret de results.

Line 235-237. I have the concern regarding the “infected mosquitoes” interpretation. I afraid the results based in the methodology cannot conclude that mosquitoes were infected. No evaluations on virus dissemination (in mosquitoes) were performed. The infection of C6/36 cells with mosquito pool extracts could be the result of the presence of virus in remaining blood of mosquitos that fed blood some days previous to be cached. In addition, it is so intriguing that flavivirus naturally infecting were identified only in mosquitoes that were not previously incriminated as flavivirus vectors (as authors mention). It had been interesting the evaluation (molecular) in well-known mosquito vectors of DENV and ZIKV pools collected in the study area.

Line 336-339. Same comment regarding infected mosquitoes, so difficult to conclude this. In addition, here is mentioned by authors that the five positive mosquito species were the most abundant species in the area. Could be this the reason to which DENV and ZIKV were detected in those mosquitoes? Please explain.

Finally, the “infecting” word in the title should be changed, may be by “detected in”.

7. PLOS authors have the option to publish the peer review history of their article (what does this mean?). If published, this will include your full peer review and any attached files.

Reviewer #3: No

Reviewer #4: No

---

## [Author Response · Author response to Decision Letter 1]

20 Jun 2020

To

Dr. Humberto Lanz-Mendoza

Academic Editor

PLOS ONE

Thank you for sending the evaluation of our manuscript. We very much appreciate the comments received from the Reviewers. We have made substantial changes to the manuscript, essentially addressing the points made by the reviewers. Accordingly, we believe that we have significantly improved the article. The changes are summarized in the reply to reviewers.

Yours sincerely,

Mauro Toledo Marrelli and coworkers

Editor Comments

1) The authors need to make clear the methodology, including positive and negative controls. 

ANSWER: IFA positive and negative controls issue was addressed. The “Detection of Flavivirus” topic was rewritten in order to avoid misunderstanding.

2) It is fundamental to determine if the mosquitoes were infected. The presence of the virus in mosquitos’ extracts is not an indicator of infection.

ANSWER: We agree with the comment. The statement was worked throughout the manuscript.

3) The authors force the conclusion that these mosquitoes might be vectors, but there is a long way to go before this can be concluded.

ANSWER: We agree with the comment. Following reviewer’s #3 suggestion, the conclusion that these mosquitoes could act as vectors for Flavivirus is too much speculative based on our data. The statement was worked throughout the manuscript and some new references were included.

Review Comments to the Author

Reviewer #3: The present study investigated the potential presence of arboviruses (more specifically flaviviruses) in wild caught mosquito species in an Environmental Reserve in Sao Paulo, Brazil. Although the findings are important for surveillance purposes, one cannot conclude that these mosquito’s species are vectors for these arboviruses. Although the authors claim they only used "non-engorged" females, there is a possibility that these mosquitoes had traces of blood in their midgut.

The authors emphasis in several instances that these mosquitoes are infected and therefore could be vectors of flaviviruses, which is not appropriate. This has to be worked throughout the manuscript.

ANSWER: Thank you for comments. We agreed with the reviewer remark, concluding that these mosquitoes could act as vectors is too much speculative based on our data. The statement was worked throughout the manuscript. 

Reviewer #4: The paper “Zika and dengue viruses infecting wild-caught mosquitoes in an environmental protection area in Brazil” describes the ZIKV and DENV infection in a C6/36 cell line, from inoculation with extracts pools (10) of wild collected mosquitoes in Brazil. Flavivirus detection was carried out by IFA and identification by qRT-PCR and sequencing. The results are interesting, but for this reviewer additional information is needed and some queries must be clarified.

Line 119. “Detection of flaviviruses” This methodology is 1/3 of the whole Materials and Methods section, but no results are provided (it is surprising because only five samples were positive). This reviewer considers it so important to the readers, mainly because the comparison between positive samples with positive and negative controls. 

ANSWER: Positive and negative controls were used only for IFA analysis as a control of the reaction per se and recommended for diagnosis proposes. The “Detection of Flavivirus” topic was rewritten in order to avoid misunderstanding. Detection of low levels of mosquito infections requires large samples (greater than 1,600 individuals) for a high probability (0.8) of detection (Gu et al. Am J Trop Med Hyg. 2004). The low detection rate obtained was included in the discussion topic.

Line 131. “…primary antibody a polyclonal anti-St Louis encephalitis virus antibody...” I have the concern about if such antibody is able to detect any flavivirus (cross-reaction), if so, a reference should be provided (Gluber et al., used a monoclonal antibody against DENV).

ANSWER: Thank you for your observation. The topic was rewritten in order to avoid misunderstanding and a recent study conducted by Cunha et al (2020) was included as reference.

Line 135. Why YFV was used as positive control? Were C6/36 infected with YFV and then performed the IFA? Again, the polyclonal anti-St Louis encephalitis virus antibody also recognize YFV (reference)? Was the IFA signal in YFV similar to problem samples?

Please clarify.

ANSWER: YFV is flavivirus, therefore able to be recognized by anti-St Louis. 17D YFV vaccine strain was used because it is a well-known standard flaviviruses strain and can be used within the established biosafety standards for flavivirus studies. C6/36 cells were inoculated with 17D YFV vaccine strain (separately from the study samples) in order to be used as a positive control for the IFA reaction. In parallel, C6/36 cells culture per se was used as negative control for the IFA reaction. This approach has been used for decades in the flavivirus routine surveillance system conducted by Vector-born Disease Laboratory. 

Line 138. “Positive flavivirus samples were analyzed...” Which positive samples (IFA)? Please clarify.

ANSWER: We clarified this issue in the text.

Line 166-167. Full genus name must be writing at first time it appears if the main body text.

ANSWER: Thank you for your observation. We corrected that throughout the text. 

Line 178. Table 1. It is not clear in the table if N (number of specimens) correspond to the amount of mosquitoes (number) or number of analyzed pools. Mainly because in the first column, the key number looks to indicate single pools. If so, there are pools from 2 to 10 mosquitos. Please clarify; it is so important to interpret de results.

ANSWER: Table 1 was rewritten in order to clarify the interpretation. 

Line 235-237. I have the concern regarding the “infected mosquitoes” interpretation. I afraid the results based in the methodology cannot conclude that mosquitoes were infected. No evaluations on virus dissemination (in mosquitoes) were performed. The infection of C6/36 cells with mosquito pool extracts could be the result of the presence of virus in remaining blood of mosquitos that fed blood some days previous to be cached. In addition, it is so intriguing that flavivirus naturally infecting were identified only in mosquitoes that were not previously incriminated as flavivirus vectors (as authors mention). It had been interesting the evaluation (molecular) in well-known mosquito vectors of DENV and ZIKV pools collected in the study area.

ANSWER: Thank you for your observation. We agreed with the reviewer remark, concluding that these mosquitoes could act as vectors is too much speculative based on our data. The statement was worked throughout the manuscript. Your suggestion to evaluate DENV and ZIKV in well-known mosquito vectors in the same surveyed area is extremely valuable and will be carefully take in consideration for future studies. 

Line 336-339. Same comment regarding infected mosquitoes, so difficult to conclude this. In addition, here is mentioned by authors that the five positive mosquito species were the most abundant species in the area. Could be this the reason to which DENV and ZIKV were detected in those mosquitoes? Please explain.

ANSWER: The most abundant mosquito species were often collected and, for sure, the more specimens sampled of a given species, the greater the chance that some viral detection will occur. However, the chances of arbovirus detection, in specially infection in mosquito populations are very low (Gu W, Novak RJ. Short report: Detection probability of arbovirus infection in mosquito populations. Am. J. Trop. Med. Hyg. 2004; 71:636-38). Furthermore, the fact of detecting infection cases is not directly related to transmission, as this phenomenon depends on the vectorial capacity of each species. This is related to conditions that extrapolate abundance, such as the rate of bites, the competence, and capacity of the pathogen to multiply in the organism of the vector, among others. So, yes, it is a possibility. However, we do not know the role of peridomestic and sylvatic mosquitoes in Flavivirus cycle in Brazil. Viral isolation from mosquitoes collected in the wild, as made here, is a rare event, and therefore our results highlight the importance of performing entomological studies and viral detection in mosquitoes.

Finally, the “infecting” word in the title should be changed, may be by “detected in”.

ANSWER: Changed.

---

## [Decision Letter · Decision Letter 2]

3 Aug 2020

PONE-D-19-34539R2

Detection of Zika and dengue viruses in wild-caught mosquitoes during field surveillance conducted in an environmental protection area, São Paulo, Brazil

PLOS ONE

Dear Dr. Marrelli,

Thank you for submitting your manuscript to PLOS ONE. After careful consideration, we feel that it has merit but does not fully meet PLOS ONE’s publication criteria as it currently stands. Therefore, we invite you to submit a revised version of the manuscript that addresses the points raised during the review process.

The manuscript requires proofread.

Please carefully review your manuscript. 

We look forward to receiving your revised manuscript.

Kind regards,

Humberto Lanz-Mendoza

Academic Editor

PLOS ONE

Additional Editor Comments (if provided):

The manuscript requires proofread.

Reviewers' comments:

Reviewer's Responses to Questions

**Comments to the Author**

1. If the authors have adequately addressed your comments raised in a previous round of review and you feel that this manuscript is now acceptable for publication, you may indicate that here to bypass the “Comments to the Author” section, enter your conflict of interest statement in the “Confidential to Editor” section, and submit your "Accept" recommendation.

Reviewer #3: (No Response)

Reviewer #4: All comments have been addressed

2. Is the manuscript technically sound, and do the data support the conclusions?

Reviewer #3: Yes

Reviewer #4: Yes

3. Has the statistical analysis been performed appropriately and rigorously? 

Reviewer #3: N/A

Reviewer #4: Yes

4. Have the authors made all data underlying the findings in their manuscript fully available?

Reviewer #3: Yes

Reviewer #4: Yes

5. Is the manuscript presented in an intelligible fashion and written in standard English?

Reviewer #3: No

Reviewer #4: Yes

6. Review Comments to the Author

Reviewer #3: Review PONE-D-19-34539_R2

Although what I have raised on my first review around the incrimination of vectors have been soften by the authors, I have the impression that, in many sections of the manuscript, they copied and pasted all the suggestions, which made the text flow very confusing and, therefore not suitable for publication in the current format. It definitely needs to go through a thorough proofreading and language review before it can be accepted.

Some examples of needed modifications/ corrections are shown below:

Rephrasing sentences on lines 21 and 23.

Line 25 Through and not throughout.

Several instances when mosquitos x mosquitoes are used.

Arbovirus is mentioned on line 50 and only explained on line 54.

Line 52-52. Please correct grammar here.

Line 212 – previously reported.

Line 225 – Plasmodium in italics.

Line 250 - Please rephrase this sentence. And has errors like “cached”.

Line 281 – species and not specie.

Line 286 – Please rephrase this sentence. Virus “are present”

Reviewer #4: In this second review, authors addressed all comments previously indicated. This reviewer consider them have done a good job by changing the title, reorganizing some data, discussion and conclusions. This reviewer consider that this new version of the manuscript is suitable for its publication.

7. PLOS authors have the option to publish the peer review history of their article (what does this mean?). If published, this will include your full peer review and any attached files.

Reviewer #3: No

Reviewer #4: No

---

## [Author Response · Author response to Decision Letter 2]

8 Sep 2020

To 

Dr. Humberto Lanz-Mendoza

Academic Editor

Plos One

Dear Dr. Lanz-Mendoza

Thank you very much for reviewing our manuscript and for the many suggestions, which have enabled us to greatly improve the text. We agree with the changes proposed by Reviewer#3 and are glad that Reviewer #4 has approved the new version of our manuscript. 

We have made the changes to the manuscript to address the points made by Reviewer#3, and we hope the article is now suitable for publication in Plos One.

Sincerely,

Mauro Marrelli

Answers: Reviewer #3

Reviewer comment: Although what I have raised on my first review around the incrimination of vectors have been soften by the authors, I have the impression that, in many sections of the manuscript, they copied and pasted all the suggestions, which made the text flow very confusing and, therefore not suitable for publication in the current format. It definitely needs to go through a thorough proofreading and language review before it can be accepted.

Answer: We thank the reviewer for pointing out these mistakes. We agree that the text was very confused and have made the necessary modifications to the manuscript in order to make it clearer and more focused. In addition, the text has been completely proofread by a native English speaker.

Reviewer comment: Some examples of needed modifications/ corrections are shown below:

Answer: We have made the modifications pointed out by the reviewer and have carried out a thorough review of the entire manuscript.

Rephrasing sentences on lines 21 and 23.

Answer: We have rephrased these sentences accordingly. Thank you for pointing this out.

Line 25 Through and not throughout.

Answer: We have corrected this.

Several instances when mosquitos x mosquitoes are used.

Answer: We have now used the term mosquitoes throughout the text. We are grateful to the reviewers for pointing this out.

Arbovirus is mentioned on line 50 and only explained on line 54.

Answer: We have modified the sentence accordingly.

Line 52-52. Please correct grammar here

Answer: We have rephrased the sentence accordingly.

Line 212 – previously reported.

Answer: We have corrected this.

Line 225 – Plasmodium in italics.

Answer: We have corrected this

Line 250 - Please rephrase this sentence. And has errors like “cached”.

Answer: We have corrected this. Thank you for pointing out this mistake.

Line 281 – species and not specie.

Answer: We have corrected this.

Line 286 – Please rephrase this sentence. Virus “are present”

Answer: We have rephrased this sentence.

---

## [Editor Report · Decision Letter 3]

30 Sep 2020

Detection of Zika and dengue viruses in wild-caught mosquitoes collected during field surveillance in an environmental protection area in São Paulo, Brazil

PONE-D-19-34539R3

Dear Dr. Marrelli,

We’re pleased to inform you that your manuscript has been judged scientifically suitable for publication and will be formally accepted for publication once it meets all outstanding technical requirements.

Kind regards,

Humberto Lanz-Mendoza

Academic Editor

PLOS ONE
---

## [Editor Report · Acceptance letter]

5 Oct 2020

PONE-D-19-34539R3 

Detection of Zika and dengue viruses in wild-caught mosquitoes collected during field surveillance in an environmental protection area in São Paulo, Brazil 

Dear Dr. Marrelli:

I'm pleased to inform you that your manuscript has been deemed suitable for publication in PLOS ONE. Congratulations! Your manuscript is now with our production department. 

Kind regards, 

on behalf of

Dr. Humberto Lanz-Mendoza 

Academic Editor

PLOS ONE